# Effect of the Soil and Ripening Stage in *Capsicum chinense* var. Jaguar on the Content of Carotenoids and Vitamins

**Julio Enrique Oney Montalvo** , **Adriana Cristina de Silva Madrigal, Manuel Octavio Ramírez Sucre** and **Ingrid Mayanin Rodríguez-Buenfil ***

Centro de Investigación y Asistencia en Tecnología y Diseño del Estado de Jalisco A.C. Sede Sureste, Tablaje Catastral 31264 Km. 5.5 Carretera Sierra Papacal-Chuburna Puerto Parque Científico Tecnológico de Yucatán, Mérida 97302, Yucatán, Mexico; juoney_al@ciatej.edu.mx (J.E.O.M.); adesilva_al@ciatej.mx (A.C.d.S.M.); oramirez@ciatej.mx (M.O.R.S.)

* Correspondence: irodriguez@ciatej.mx

**Abstract:** The purpose of this work was to investigate the effect of the ripening stage and type of soil on the concentration of carotenoids and vitamins in Habanero pepper (*Capsicum chinense* Jacq.). Pepper plants were grown in two soils named according to the Mayan classification as: *K'ankab lu'um* (red soil) and *Box lu'um* (black soil). The results of two harvests at 320 and 334 PTD (post-transplant day) showed that the ripening stage exhibited a significant effect ($p < 0.05$) on the concentration of carotenoids and vitamins, while the effect of the soil type was negligible. The concentration of carotenoids decreases as the ripening process of the fruit takes place, with the highest concentration of lutein ($49.47 \pm 0.34$ mg/100 g of dry mass), β-carotene ($99.92 \pm 0.69$ mg/100 g of dry mass) and β-cryptoxanthin ($20.93 \pm 0.04$ mg/100 g of dry mass) in the unripe peppers. The concentration of vitamins increases as the ripening process develops, with the highest concentration of Vitamin E ($9.69 \pm 0.02$ mg/100 g of dry mass) and Vitamin C ($119.44 \pm 4.72$ mg/100 g of dry mass) in the ripe peppers. This knowledge could be used to select the best ripening stage to harvest Habanero peppers according to the use of the pepper and to the needs of producers/company.

**Keywords:** *Capsicum chinense*; total carotenoids; Vitamin C; Vitamin E; lutein; β-cryptoxanthin; β-carotene

## 1. Introduction

The Habanero pepper (*Capsicum chinense* Jacq.) var. Jaguar is a fruit of a species of the Solanaceae family used as an important ingredient in the gastronomy of countries such as Mexico, China, Thailand and South Korea, and it is known around the world for its high pungency [1]. *Capsicum chinense* Jaq. var. Jaguar tends to grow in clay soils and is perfectly adapted to the adverse effects of tropical environmental conditions, such as those presented on the Yucatan Peninsula (high temperature and humidity), where the Habanero pepper has a significant economic and social impact [2,3]. The life cycles of pepper plants include four phenological stages: (1) vegetative, (2) flowering, (3) fruiting and (4) production, depending on the climate and growing conditions. Vegetative stage occurs approximately at 50 post-transplant days (PTD), flowering stage normally happens between 70 and 85 PTD, fruiting stage starts at 100 PTD, and finally, production stage starts approximately at 120 PTD [2–4].

Habanero pepper fruits var. Jaguar are usually harvested in a period of three to seven months in open field and up to two years in the greenhouse. These fruits are characterized by a length of 3.8 to 5.5 cm, a diameter of 2.5 to 3 cm, a weight of 6.5 to 10 g, and with an average firmness of 58.3 N cm$^{-2}$. Pepper plants produce unripe green fruits, which turn orange when fully ripe, with an intermediate stage of half ripe peppers with sections of both colors [3]. The green color of unripe fruits is mainly attributed to the chlorophyll content and pigments as carotenoids, while the orange color is caused by a decrease in the

content of those compounds and the synthesis of other metabolites (other carotenoids or polyphenols) [5].

These peppers are considered a rich source of important metabolites with potential health-promoting properties, with capsaicinoids being the main metabolites; however, other metabolites such as carotenoids and vitamins are important to the food and pharmaceutical industry due to their health benefits [6]. Carotenoids are a group of fat-soluble pigments widely distributed in nature [7]. These compounds have been studied for their antioxidant effects that are associated with their ability to prevent cancer; therefore, the consumption of foods rich in carotenoids is recommended, such as Habanero pepper [8]. Lutein, β-cryptoxanthin and β-carotene are considered characteristic carotenoids of Habanero peppers, reporting concentrations of 37.5 to 62.7 mg/100 g of dry mass, respectively [9,10].

On the other side, vitamins are a group of organic compounds that act as coenzymes in different metabolic processes, such as the metabolism of fatty acids, amino acids, and purines [11]. The main vitamins reported in peppers of *Capsicum* genus are ascorbic acid (Vitamin C) and α-tocopherol (Vitamin E) [12,13]. In these peppers, Vitamin C exhibit concentrations ranging from 43 to 247 mg/100 g of fresh pepper, contributing with 50 to 100% of the daily requirements (daily intake recommended: 15 mg for children and 90 mg for adults, according to the USDA) [12]. Vitamin E is presented as four vitamers (molecules with similar vitamin activity): α-tocopherol, β-tocopherol, γ-tocopherol and δ-tocopherol. The α-tocopherol is the most important because it shows: (1) the highest vitamers proportion and (2) the highest vitamin activity. In fresh Habanero peppers the concentration of Vitamin E ranges from 2 to 17 mg/100 g [14].

The concentration of carotenoids and vitamins in Habanero peppers depends on different factors, the most important are: ripening stage, temperature, solar radiation and type of soil [15]. In the case of carotenoids, the aforementioned factors mainly generate an effect on the enzyme phytoene synthetase. This enzyme is considered a limiting factor in the speed of carotenoid biosynthesis because its expression or activity will alter the rate of production of carotenoids [16]. The factors that affect the biosynthesis of vitamin C the solar radiation to which the fruit and the plant are exposed and ripening stage, because it is related to the amount of sugars developed in the pepper [17]. Moreover, it has been reported that the main factors that affect the biosynthesis of Vitamin E are temperature, solar radiation to which the crop is subjected, as well as the ripening stage of the fruit [18,19].

In addition, the concentration of metabolites in Habanero peppers is impacted by physicochemical characteristics of the soils where the cultivation is carried out [20]. The main soils used in Yucatan Peninsula are: *K'ankab lu'um* (red soil) and *Box lu'um* (black soil) [20]. This classification comes from the Mayan language, and is widely used in the Yucatan Peninsula (southeast region in Mexico) to refer to soils of different colors [21]. The work previously conducted by Oney-Montalvo et al. [22] reported that the red soils have the highest concentration of calcium ($2075.28 \pm 29.70$ mg kg$^{-1}$), magnesium ($779.23 \pm 12.71$ mg kg$^{-1}$), phosphorus ($11.00 \pm 2.38$ mg kg$^{-1}$), iron ($6.27 \pm 0.21$ mg kg$^{-1}$) and the lowest concentration of organic matter ($5.16 \pm 0.04$%). In contrast, black soils have a higher concentration of nitrogen ($52.01 \pm 7.05$ mg kg$^{-1}$), organic matter ($10.93 \pm 0.23$%) and manganese ($5.24 \pm 0.45$ mg kg$^{-1}$) [22].

Based on the above, the goal of the current study was to evaluate the content of carotenoids (lutein, β-carotene and β-cryptoxanthin) and vitamins (ascorbic acid and α-tocopherol) during the ripening process of the Habanero pepper (*Capsicum chinense* Jacq.), in order to understand soil's contribution to the production of these metabolites and find the best ripening stage with the highest concentration of these metabolites. This knowledge would be useful to add value to this fruit due to the nutraceutical characteristics provided by vitamins and carotenoids.

## 2. Materials and Methods

### 2.1. Growing Conditions of the Habanero Pepper

The cultivation of Habanero peppers (*Capsicum chinense* Jacq. 'Jaguar') started on January 18th, 2019. For this, plantlets of Habanero pepper were used with 48 days of germination and transplanted into polyethylene bags containing 12 kg of the two types of soil from Yucatan (Figure 1): red soil (*K'ankab lu'um*) and black soil (*Box lu'um*). Harvests were carried out at 320 and 000 334 PTD (post-transplant day) in order to observe the changes caused by the life cycle of the plant. The harvest dates were selected according to the availability of peppers and to obtain the peppers in the three different ripening stages: unripe, half ripe and ripe (Figure 1). The plants were established in a greenhouse in Sierra Papacal, Yucatán in Mexico (CIATEJ Sede Sureste, Mérida, Yucatán, México). It had a north–south orientation, with a roof of triple-layer plastic cover (25% shade), and side walls of high-density anti-trip plastic screens. Water with an electrical conductivity from 2.8 to 3.4 mS was used for irrigation. To monitor temperature, light, and relative humidity in the crop, Data Loggers were placed throughout the greenhouse. The results are presented in Table A1 of the Appendix A.

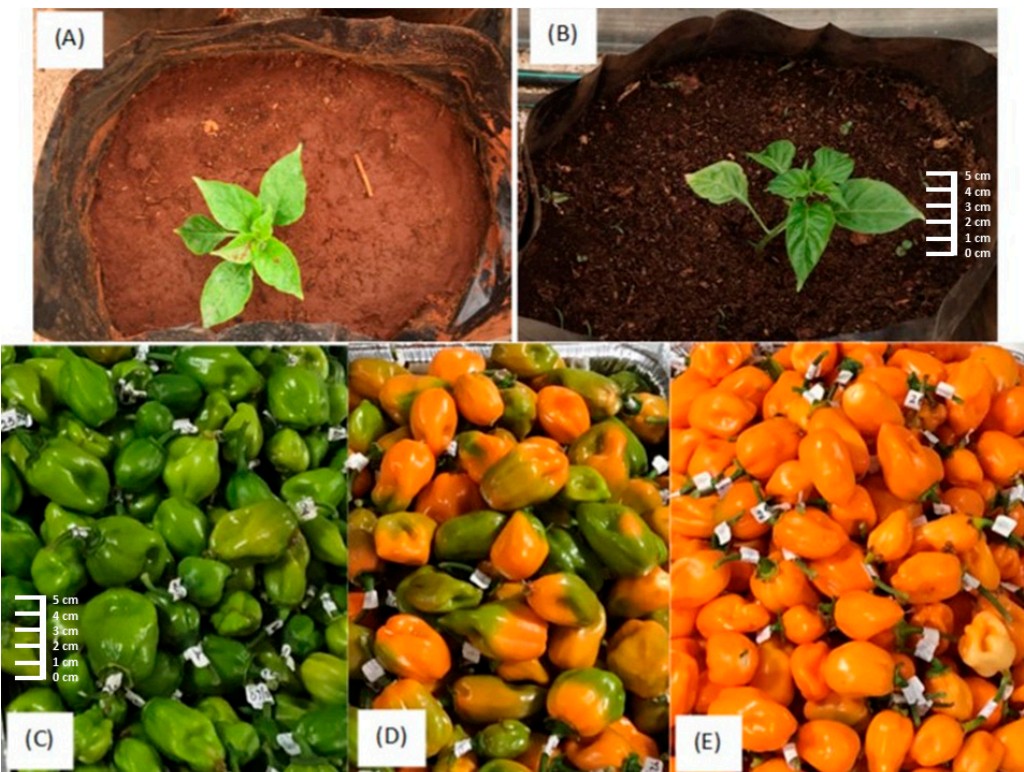

**Figure 1.** Types of soil used (**A**) red soil, (**B**) black soil, and maturity stages of the analyzed fruits (**C**) unripe (green pepper), (**D**) half ripe (green-orange pepper) and (**E**) ripe (orange pepper).

For fertilization, the methodology of Medina-Lara et al. [23] was used, which is recommended for Habanero peppers cultivated in soils of Yucatan to develop an adequate production of fruits. Fertilizer Triple 18 Ultrasol® (SQM, Santiago de Chile, Chile) composed of nitrogen, phosphorus, and potassium at a concentration of 18% was used after 10 post-transplant days (PTD). The fertilizer was applied by irrigation with water (1 L) twice a week. The micronutrients (B, Ca, Cu, Co, Fe, Mn, Mo, Mg and Zn) were sprayed on the leaves once a week with commercial product Bayfolan® Forte (Bayer CropScience, Mexico City, Mexico) diluting 24 mL in 16 L of water. After 20 PTD and before floral initiation, a growth regulator containing gibberellin, cytokinin and auxin (Biozyme®-TF, Arysta LifeScience, Guatemala City, Guatemala) was sprayed (16 mL diluted in 16 L of water) once a week. Irrigation was applied twice a week during the first 15 days after

post-transplant; subsequently, the irrigation frequency was maintained at 2 L per bag, every third day.

## 2.2. Sample Preparation

Habanero peppers were collected and classified by soil (red and black), ripening stage (unripe or green color, half ripe or green–orange color and ripe or orange color) and harvest time (320 and 334 PTD). Approximately 100 peppers were collected by each combination. In addition, the peppers were selected according to the criteria of the Mexican standard NOM-189-SCFI-2017, which indicates that the Habanero peppers must be bell shaped with a pointed end and be entire, without stains, diseases or pests. The selection of the peppers was made by visual inspection according to color and other characteristics required by the regulations. After collected, the peppers were lyophilized according to the methodology reported by Oney-Montalvo et al. [24]. Lyophilization was carried out at a temperature of $-50\ °C$ and a pressure of 0.200 mBar for 72 h. Then, the peppers were ground with a mortar and pestle and, finally, sieved on # 35 mesh (500 μm). The resulting powder was stored for one week in a plastic bag at room temperature (25 °C) and protected from light with aluminum foil.

## 2.3. Extraction of Carotenoids

The extraction of the carotenoids was carried out using as a reference to the work carried out by Ye et al. [25]. Approximately 500 mg of Habanero pepper powder were weighed and mixed with 2.5 mL of ethanol in a falcon tube. The mix was sonicated at 42 KHz for 20 min at room temperature and then, centrifuged at 3500 rpm for 30 min. The supernatant was collected and filtered through a 0.2 μm PTFE filter and analyzed by Ultra-Performance Liquid Chromatography (UPLC).

## 2.4. Extraction of Vitamins

To extract the ascorbic acid (Vitamin C) the methodology reported by Oney-Montalvo et al. [24] was used. Fifty mg of Habanero pepper powder were weighed and mixed with 4 mL of water:acetonitrile (80:20) in a falcon tube. The mixture was sonicated at 42 KHz for 30 min at room temperature and then, filtered through a 0.2 μm PTFE filters and analyzed by UPLC. The extraction of α-tocopherol (Vitamin E) was carried out with the same extraction methodology of carotenoids.

## 2.5. Total Carotenoids by Spectrometry UV-VIS

Five hundred milligrams of dry Habanero pepper powder were weighed and mixed with 2.5 mL of methanol, then sonicated for 20 min and centrifuged at 3500 rpm at 4 °C for 20 min. The extract obtained was filtered and diluted with distilled water in a 1:1 ratio and read at 450 nm in a spectrometer (Jenway, United Kingdom). The calibration curve was prepared at 5 concentrations (0.075, 0.1, 0.2, 0.3 and 0.4 mg mL$^{-1}$) from a stock solution of β-carotene 0.5 mg mL$^{-1}$ in methanol.

## 2.6. Determination of Antioxidant Activity

Antioxidant activity was determined by inhibition of the 2,2-diphenyl-1-picrylhydrazyl (DPPH) radical [26]. The extracts used were obtained from the extraction of total carotenoids. The DPPH solution was prepared with HPLC grade methanol and diluted to a solution with an absorbance of $0.7 \pm 0.002$ at 515 nm. Subsequently 100 μL of the pepper extract was added to 3.9 mL of DPPH solution, the mixture was stirred, allowed to stand 30 min and read in the spectrophotometer (Jenway, UK) at 515 nm. The percentage of DPPH was calculated using the following formula:

$$\%\text{DPPH} = \left( \frac{A_{\text{control}} - A_{\text{Sample}}}{A_{\text{control}}} \right) \times 100$$

where $A_{Control}$ is the absorbance of the control (0.7 Abs) and $A_{Sample}$ is the absorbance of the sample.

*2.7. Chromatographic Analysis of Carotenoids*

Analysis of metabolites (individual carotenoids and vitamins) was performed in an UPLC Acquity H Class (Waters, Milford, MA, USA) with a diode array detector and an Acquity UPLC HSS C18 column (100 Å, 1.8 μm, 2.1 × 50 mm) (Waters, Milford, MA, USA). Validation of the chromatographic methods was carried in accordance with the *Eurachem Guide: The Fitness for Purpose of Analytical Methods—A Laboratory Guide to Method Validation and Related Topics* [27] reported in a previous work conducted by Chel-Guerrero et al. [28].

Quantification of carotenoids was carried out using the method established by Delpino-Rius et al. [29] as reference. It consists of the following conditions: flow speed of 0.5 mL min$^{-1}$ with a column temperature set at 35 °C and injection volume of 2 μL. Detection was performed at a wavelength of 475 nm. The carotenoids were separated using acetonitrile:methanol (70:30) as mobile phase. These were quantified by external calibration prepared with 3 carotenoid standards from Sigma Aldrich® (Toluca, Mexico) analytical standard grade: lutein (purity ≥ 96%), β-carotene (purity ≥ 95%) and β-cryptoxanthin (purity ≥ 95%). First, a stock solution at a concentration of 0.5 mg mL$^{-1}$ was prepared of all standards, and then the calibration curve was prepared in the range of 0.001 to 0.1 mg mL$^{-1}$. Carotenoids were identified in the samples with a comparison with the retention time of the standards. The chromatogram obtained from the analysis of carotenoids in the Habanero pepper using the method described is represented in Figure A1 in Appendix B.

*2.8. Chromatographic Analysis of Vitamins*

Ascorbic acid was determinate with the method established by Oney-Montalvo et al. [24]. The mobile phase was composed of formic acid 0.1% in water with a flow speed of 0.25 mL min$^{-1}$, a column temperature set at 27 °C and an injection volume of 2 μL. The detection was performed at wavelength of 244 nm. This was quantified by external calibration prepared in the range of 0.5 to 5 μg mL$^{-1}$ with ascorbic acid analytical grade standard (purity ≥ 99%) from Sigma Aldrich® (Toluca, Mexico).

The α-tocopherol was determined using the method developed by Gledhill [30]. They used formic acid at 0.2% in acetonitrile:methanol (50:50) as mobile phase at a flow speed of 0.5 mL min$^{-1}$ with a column temperature set at 35 °C and an injection volume of 2 μL. Detection was performed at a wavelength of 290 nm. Quantification was made by external calibration curve prepared in the range of 1.5 to 75 μg mL$^{-1}$ with α-tocopherol analytical grade standard (purity ≥ 96) from Sigma Aldrich® (Toluca, Mexico). Both, ascorbic acid and α-tocopherol were identified in samples with a comparison with the retention time of the standards. The chromatograms obtained from the analysis of vitamins in Habanero pepper using the method previously described are presented in Figures A2 and A3 in Appendix B.

*2.9. Statistical Analysis*

Analysis of variance (ANOVA) was used with $\alpha = 0.5\%$ to analyze all data. The results were evaluated by descriptive and dispersion statistics, being the values presented the mean ± standard deviation. The statistical test used for separation of means was the least significant difference with a confidence level of 95%. All statistical analysis were performed with the software Statgraphics Centurion XVII.II-X64 (Statgraphics Technologies Inc., Plains, VA, USA).

**3. Results**

*3.1. Quantification of Carotenoids and Vitamins by UPLC*

The results obtained from the quantification of vitamins (ascorbic acid and α-tocopherol) and carotenoids (lutein, β-carotene and β-cryptoxanthin) by UPLC-DAD are presented in Table 1. The highest concentration of carotenoids was quantified in unripe peppers grown

in red soil and harvested at 334 PTD, concentrations of 49.47 ± 0.34 mg/100 g of lutein, 99.92 ± 0.69 mg/100 g of β-carotene and 20.93 ± 0.04 mg/100 g of β-cryptoxanthin were reported. In the chromatograms (C) and (D) of Figure A1 in Appendix B it is observed that the signals corresponding to the peaks of the carotenoids of the unripe pepper show a higher intensity compared to the ripe pepper. On the other hand, the highest concentration of vitamins was quantified in ripe pepper (orange) harvested at 320 PTD, reporting a tocopherol concentration of 9.69 ± 0.02 mg/100 g in peppers grown in the black soil. In the chromatograms (B) and (C) of Figure A3 of Appendix B it is observed that the peak corresponding to Vitamin E is greater in the ripe pepper compared to the unripe. Ascorbic acid had a concentration of 119.44 ± 4.72 mg/100 g in peppers grown in the red soil. Additionally, a higher peak of ascorbic acid corresponded to the ripe pepper (chromatogram (B) and (C) of Figure A2 of Appendix B).

**Table 1.** Vitamins and carotenoids in unripe (green pepper), half ripe (green-orange pepper) and ripe (orange pepper) Habanero pepper (*Capsicum chinense* Jacq.) cultivated in different soils at two PTD.

| | | | Vitamins | | Carotenoids | | |
|---|---|---|---|---|---|---|---|
| PTD | Soil | Ripening Stage | Vitamin C (mg/100 g) | Vitamin E (mg/100 g) | Lutein (mg/100 g) | β-Carotene (mg/100 g) | β-Cryptoxanthin (mg/100 g) |
| 320 | Red | Unripe | 72.44 ± 1.74 [e] | 4.24 ± 0.01 [g] | 29.13 ± 0.04 [c] | 63.59 ± 0.36 [c] | 14.65 ± 0.10 [b] |
| | | Half ripe | 86.01 ± 3.72 [c] | 8.00 ± 0.04 [c] | 10.22 ± 0.15 [f] | 12.39 ± 0.36 [f] | 2.56 ± 0.04 [d] |
| | | Ripe | 119.44 ± 4.72 [a] | 8.96 ± 0.03 [b] | 1.43 ± 0.06 [i] | 0.49 ± 0.00 [i] | 1.44 ± 0.05 [e] |
| | Black | Unripe | 66.61 ± 3.85 [f] | 3.67 ± 0.01 [h] | 25.43 ± 0.37 [d] | 56.08 ± 0.35 [d] | 10.76 ± 0.15 [c] |
| | | Half ripe | 73.43 ± 1.88 [e] | 7.06 ± 0.03 [e] | 7.43 ± 0.29 [g] | 7.30 ± 0.36 [g] | 1.52 ± 0.01 [e] |
| | | Ripe | 103.89 ± 3.44 [b] | 9.69 ± 0.02 [a] | 1.55 ± 0.06 [i] | 0.00 ± 0.00 [i] | 1.54 ± 0.04 [e] |
| 334 | Red | Unripe | 67.99 ± 0.79 [f] | 5.57 ± 0.01 [f] | 49.47 ± 0.34 [a] | 99.92 ± 0.69 [a] | 20.93 ± 0.04 [a] |
| | | Half ripe | 70.51 ± 4.27 [e] | 7.40 ± 0.01 [d] | 14.78 ± 0.14 [e] | 16.71 ± 0.00 [e] | 10.62 ± 0.02 [c] |
| | | Ripe | 77.99 ± 0.45 [d] | 8.96 ± 0.01 [b] | 1.97 ± 0.02 [h] | 1.99 ± 0.00 [h] | 14.85 ± 0.05 [b] |
| | Black | Unripe | 48.36 ± 0.01 [g] | 5.66 ± 0.01 [f] | 43.95 ± 0.36 [b] | 82.63 ± 1.78 [b] | 19.34 ± 0.36 [a] |
| | | Half ripe | 51.27 ± 0.37 [g] | 6.94 ± 0.04 [e] | 14.09 ± 0.11 [e] | 15.52 ± 0.71 [e] | 10.23 ± 0.30 [c] |
| | | Ripe | 72.91 ± 0.79 [e] | 9.44 ± 0.19 [a] | 2.15 ± 0.19 [h] | 1.76 ± 1.07 [hi] | 15.08 ± 0.02 [b] |

Note: Values are expressed in average concentration ± standard deviation (number of replications = 3). Different letters in the same column indicate statistically significant differences using least significant difference (LSD) test at $p \leq 0.05$.

### 3.2. Total Carotenoids and Antioxidant Activity by Spectroscopy

The results in Table 2 show the highest concentration of total carotenoids identified in unripe peppers (green) with a concentration of 722.19–735.31 mg/100 g at 320 PTD on black soil and at 334 PTD on red soil, respectively. In contrast, a high antioxidant activity was shown in half ripe peppers grown in black soil and harvested at 320 and 334 PTD (89.21 ± 0.22–91.12 ± 0.06%, respectively); the ripe peppers grown in red soil and harvested at 334 PTD also presented high activity (89.49 ± 0.05%).

### 3.3. Statistical Analysis

*p* Values showed that the ripening stage had a significant effect ($p < 0.05$) on all the response variables studied (Table 3). The soil variable and the interaction between the ripening stage and type of soil did not have a significant effect in any of the response variables studied. On the other hand, PTD showed significant effect on all the metabolites evaluated except for the α-tocopherol ($p = 0.0896$).

**Table 2.** Total carotenoids and antioxidant activity (DPPH) in unripe (green pepper), half ripe (green-orange pepper) and ripe (orange pepper) Habanero pepper (*Capsicum chinense* Jacq.) cultivated in different soils.

| PTD | Soil | Ripening Stage | Antioxidant Activity (%) | Total Carotenoids (mg/100 g) |
|---|---|---|---|---|
| 320 | Red | Unripe | 86.42 ± 0.21 [d] | 402.31 ± 0.62 [g] |
| | | Half ripe | 87.18 ± 0.10 [c] | 373.01 ± 0.50 [h] |
| | | Ripe | 84.43 ± 0.15 [e] | 294.19 ± 0.75 [i] |
| | Black | Unripe | 86.18 ± 0.10 [d] | 722.19 ± 0.84 [a] |
| | | Half ripe | 91.12 ± 0.06 [a] | 545.09 ± 1.10 [d] |
| | | Ripe | 80.01 ± 0.06 [f] | 559.73 ± 1.71 [d] |
| 334 | Red | Unripe | 87.82 ± 0.17 [c] | 735.31 ± 0.84 [a] |
| | | Half ripe | 88.14 ± 0.17 [bc] | 633.42 ± 1.31 [b] |
| | | Ripe | 89.49 ± 0.05 [ab] | 474.68 ± 0.76 [e] |
| | Black | Unripe | 85.90 ± 0.10 [d] | 600.40 ± 0.87 [c] |
| | | Half ripe | 89.21 ± 0.22 [ab] | 473.59 ± 0.50 [e] |
| | | Ripe | 88.07 ± 0.21 [bc] | 446.91 ± 1.24 [f] |

Note: Values are expressed in average ± standard deviation (number of replications = 3). Different letters in the same column indicate statistically significant differences using least significant difference (LSD) test at $p \leq 0.05$.

**Table 3.** *p* Values of the different factors evaluated and their respective interactions for each of the quantified carotenoids and vitamins.

| | A: Ripening Stage | B: Soil | AxB | PTD |
|---|---|---|---|---|
| Lutein | <0.0001 * | 0.2991 | 0.6118 | 0.0004 * |
| β-cryptoxanthin | <0.0001 * | 0.1175 | 0.2195 | <0.0001 * |
| β-carotene | <0.0001 * | 0.1234 | 0.3143 | 0.0009 * |
| Vitamin C | <0.0001 * | 0.9946 | 0.2075 | <0.0001 * |
| Vitamin E | <0.0001 * | 0.5503 | 0.0928 | 0.0896 |
| Total carotenoids | 0.0204 * | 0.1457 | 0.6010 | 0.0999 |
| Antioxidant activity (DPPH) | 0.0089 * | 0.5315 | 0.4010 | 0.0117 * |

Note: * Significant effect; AxB = Interaction of ripening stage with soil; PTD = post-transplant days.

### 3.4. Analysis of the Ripening Stage Effect

The effect of ripening stage on the concentration of vitamins and carotenoids in the Habanero pepper fruit was analyzed to understand how this factor affects the concentration of those important metabolites. Figure 2 shows the average results of antioxidant activity (Figure 2A) and total carotenoids by spectroscopy (Figure 2B). The concentration of total carotenoids decreases according to the ripening stage of the fruit, the unripe peppers presented the highest concentration (615.05 ± 154.29 mg/100 g of dry mass). On the other hand, the antioxidant activity behaves with a maximum in half ripe peppers (88.91 ± 1.69%).

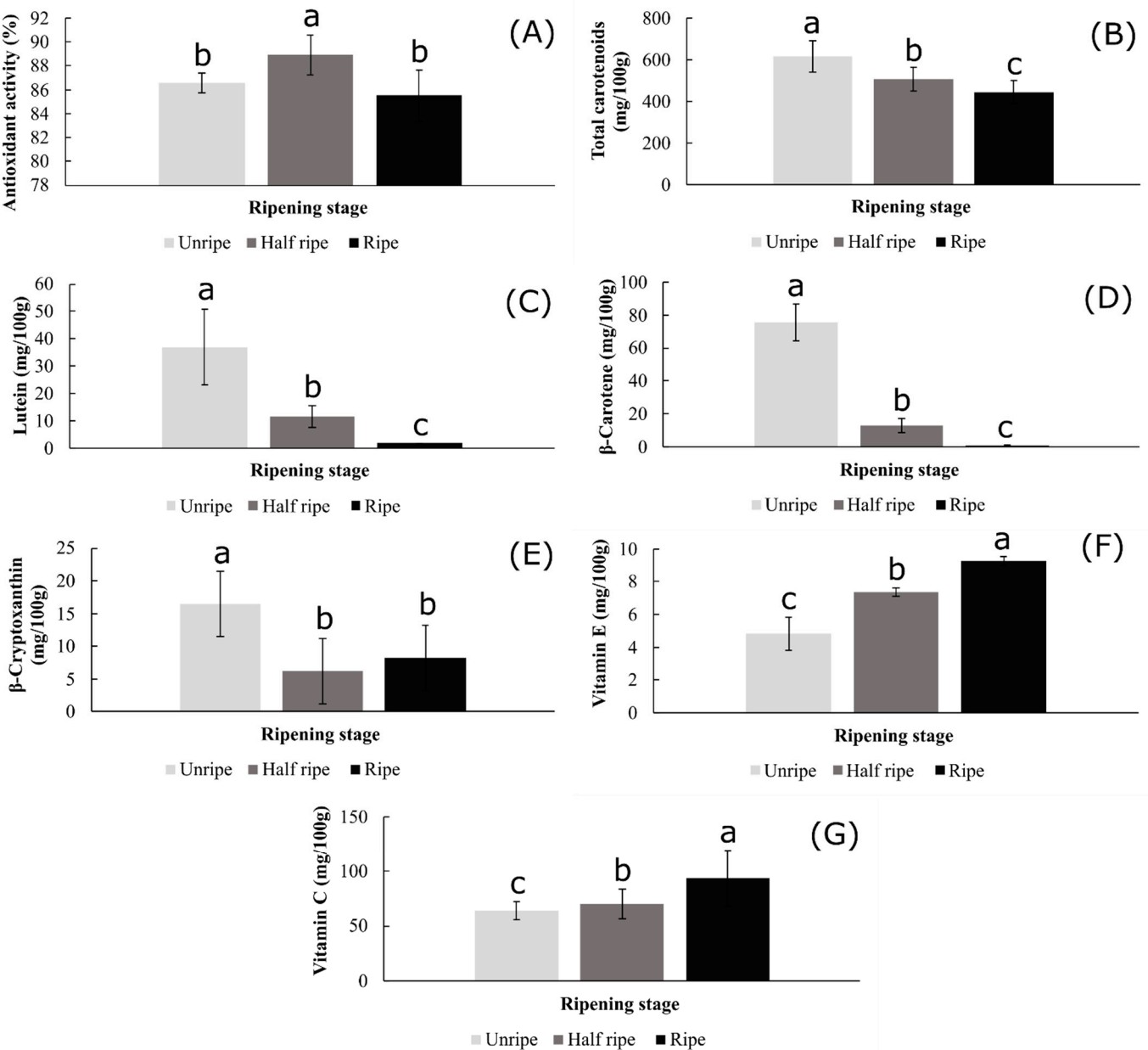

**Figure 2.** Determination of (**A**) antioxidant activity, (**B**) total carotenoids, (**C**) lutein, (**D**) β-carotene, (**E**) β-cryptoxanthin, (**F**) Vitamin E (α-tocopherol) and (**G**) Vitamin C (ascorbic acid) by ripening stage. Data are presented as means (mg/100 g of dry mass) and error bars represent standard deviation (number of replications = 3). Different letters indicate statistically significant differences using least significant difference (LSD) test at $p \leq 0.05$.

Figure 2 shows the average concentration of carotenoids and vitamins at different ripening stages. It was evident that the concentration of lutein (Figure 2C), β-carotene (Figure 2D) and β-cryptoxanthin (Figure 2E) decreased with the ripening process of the Habanero pepper. Therefore, the highest concentration of lutein (37.00 ± 13.74 mg/100 g of dry mass), β-carotene (75.56 ± 22.23 mg/100 g of dry mass) and β-cryptoxanthin (16.42 ± 5.25 mg/100 g of dry mass) was found in the unripe pepper. The concentration of α-tocopherol (Figure 2F) and ascorbic acid (Figure 2G) increased with the ripening process of the Habanero pepper. The results show an increase in the two vitamins analyzed as the ripening process progresses. Ripe fruit showed the highest concentration of Vitamin E (9.26 ± 0.09 mg/100 g of dry mass) and Vitamin C (93.56 ± 25.06 mg/100 g of dry mass), while the lowest concentration was obtained in the unripe pepper.

## 4. Discussion

A similar behavior was found for total and individual carotenoids conducted by spectroscopy and chromatography: the concentration decreased in both cases as the ripening process proceeds. On the other hand, the antioxidant activity determined by DPPH showed the highest antioxidant activity in half ripe Habanero peppers (green–orange). This behavior could be caused by other metabolites with antioxidant activity that are found in higher concentration on Habanero pepper, for example: vitamins, capsaicinoids and polyphenols [22,31]. The antioxidant activity reported in the present work was in the range of 80.01–91.12%; Segura-Campos et al. [32] reported results of 56.29 to 94.98% in Habanero pepper (*Capsicum chinense* Jacq.) using the ABTS radical scavenging assay. This is type of assay for the determination of antioxidant activity that has been correlated with DPPH assay; both methodologies has been used to determine mainly the antioxidant activity by phenolic compounds of fruits, vegetables and other plant products [33]. This result confirms the antioxidant potential of the pepper.

The ripening stage had a significant difference in the bioactive compounds studied in the present work (carotenoids and vitamins). This could be associated with the changes in gene expression that the fruit undergoes during the ripening process [34]. These changes affect the synthesis of enzymes that play an important role in the synthesis pathway of these compounds, for example: The L-galactono-1, 4-lactone dehydrogenase in the case of ascorbic acid and the phytoene synthetase for carotenoids [17,35]. In contrast, the case of α-tocopherol may be associated with an increase in the expression of the VTE1 gene, responsible for the synthesis of the enzyme tocopherol cyclase, which produces γ-tocopherol, a precursor of α-tocopherol [19]. Moreover, the soils used for cultivation have different chemical composition that has been shown a significant effect on the production of polyphenols and capsaicinoides [22,36], however, this factor did not show a significant effect on the concentration of the metabolites studied in the present work. This might be because the predominant effect of the ripening stage could mask some possible effect of the soil. On the other hand, the effect of PTD on the production of carotenoids and vitamins may be caused by the life cycle of the plant, a factor that is associated with the absorption of nutrients, the synthesis and the distribution of secondary metabolites (for example capsaicinoids) [23]. This behavior is similar to changes previously found in concentrations of capsaicinoids and polyphenols of Habanero peppers at different PTDs [22,36]. Furthermore, environmental changes (temperature, humidity, and lighting) at different PTDs could affect the production of enzymes involved in the synthesis of vitamins and carotenoids [16–18] (Table A1 of Appendix A).

The unripe peppers had the highest content of the three evaluated carotenoids (lutein, β-cryptoxanthin and β-carotene). This behavior was also reported by Gómez-García and Ochoa-Alejo [15]. They demonstrated that the concentration of these carotenoids decreases during the ripening process of the *Capsicum* spp., because these compounds are bio-transformed by the carotenoid synthesis pathway, obtaining xanthophylls. These compounds are considered products derived from the oxidation of carotenoids and are responsible for the characteristic orange color of the Habanero pepper [35]. In the present work, the maximum concentration of lutein (49.4 mg/100 g of dry mass) was quantified in the unripe peppers, a value close to that reported by Troconis-Torres et al. [10] (37.5 ± 1.2 mg/100 g). In the case of β-carotene, the highest concentration obtained was developed by the unripe peppers (99.92 mg/100 g of dry mass), similar to that reported by Howard et al. [37] (86.1 ± 3.8 mg/100 g of dry mass). In addition, the highest content of β-cryptoxanthin obtained in the present study (20.93 ± 0.04 mg/100 g of dry mass) was similar to that reported (1.9–35.3 mg/100 g of dry mass) by Giuffrida et al. [9] and Howard et al. [37].

The highest concentration of ascorbic acid was found in ripe peppers, this behavior was demonstrated by other works. For example, Segura-Campos et al. [32] also quantified the highest concentration of ascorbic acid in ripe Habanero peppers reporting a high concentration (281.73 mg/100 g of dry mass). The higher concentration of ascorbic acid is

related to the ripening stage of the fruit and to the amount of sugars that are precursors to produce this vitamin [17]. In addition, the expression of the gene that synthesize the enzyme L-galactono-1,4-lactone dehydrogenase is another factor that affects the concentration of ascorbic acid. This enzyme is responsible for producing ascorbic acid and its concentration increase with the ripening stage [17]. The ascorbic acid concentration quantified in the present work (119.44 ± 4.72 mg/100 g of dry mass) is in the range reported by Topuz and Ozdemir [38] and Bae et al. [39] (75.1–230 mg/100 g of dry mass).

The previous work carried out by Menichini et al. [40] reported a higher concentration of Vitamin E in ripe Habanero pepper (5.90 ± 0.24 mg/100 g of dry mass) compared to the unripe (2.77 ± 0.18 mg/100 g). The ripening stage is the factor that exhibited the major effect on the concentration of Vitamin E, obtaining the highest concentrations when the pepper is ripe (orange). This behavior was also observed by Wahyuni et al. [14]. They reported that the concentration of this vitamin increases proportionally to the ripening stage in fruits of the *Capsicum* genus. It can be inferred that the ripening stage is related with a higher concentration of the vitamin, considering its biosynthesis pathway. On the other hand, the Vitamin E quantified in the present work (9.69 ± 0.02 mg/100 g of dry mass) was higher than the reported by Saha et al. [41], who found a concentration of 4.39 ± 0.11 mg/100 g of dry mass; this might be due to the use of another variety of pepper (*Capsicum* spp.).

The changes in the concentration of vitamins and carotenoids between the results obtained in the present work and those reported in the literature, could be associated with environmental characteristics (temperature, humidity, and luminosity) or factors that affect the synthesis of these metabolites [17,42]. The environmental conditions prior to the Habanero pepper harvests are reported in Table A1 of the Appendix A. These measurements show that on both harvests' times a relative humidity above 85% was obtained, as well as temperatures above 35 °C and an illumination greater than 4000 cd/m$^2$. Although, low temperatures have been shown to contribute to the synthesis of some carotenoids such as: lutein, violaxanthin and antheraxanthin, high temperatures have not been reported to negatively influence carotenoid production in fruits [43]. On the other hand, the illumination generated by solar radiation has shown important effects to the synthesis of these compounds [42]. In the case of vitamins, literature reports an increase in vitamin production caused by high temperatures and prolonged exposure to solar radiation. This could cause higher concentrations in the present work, higher than those reported by Menichini et al. [40] and Saha et al. [41].

The present work contributes to increase the knowledge of the effect of the type of soil and ripening stage on the concentration of carotenoids and vitamins in the Habanero pepper. The degree of ripening had a significant effect on the concentration of these metabolites, while type of soil was negligible; the degree of ripening is an important factor to consider according to the characteristics that are desired in the fruit.

## 5. Conclusions

The results indicate that the concentration of carotenoids and vitamins in the Habanero pepper (*Capsicum chinense* Jacq.) change due to the effect of the ripening stage, while the effect of the soil was negligible. The content of total carotenoids decreased with the ripening of the fruit, exhibiting the highest concentration in unripe peppers (615.05 ± 154.29 mg/100 g of dry mass). Major concentrations of lutein (49.47 ± 0.34 mg/100 g of dry mass), β-carotene (99.92 ± 0.69 mg/100 g of dry mass) and β-cryptoxanthin (20.93 ± 0.04 mg/100 g of dry mass) were registered in the unripe pepper. On the other hand, ripe peppers showed the highest concentration of Vitamin E (9.69 ± 0.02 mg/100 g of dry mass) and Vitamin C (119.44 ± 4.72 mg/100 g of dry mass), while the maximum antioxidant activity was found in half ripe peppers (88.91 ± 1.69%). The knowledge obtained could be used by the agribusiness to select the best ripening stage necessary in the Habanero pepper for its later use (if a high content of carotenoids or vitamins is desired). Habanero pepper extracts or powders could be used to fortify foods or supplements, or to develop new functional

foods, providing added value to this fruit. On the other hand, this study also provides information about similar results obtained with the two types of soils used. Therefore, both could be used for plant growth when the main interest is to obtain fruits with a high content of carotenoids or vitamins.

**Author Contributions:** Conceptualization, J.E.O.M. and I.M.R.-B.; methodology, J.E.O.M., I.M.R.-B. and A.C.d.S.M.; software, A.C.d.S.M.; validation, I.M.R.-B. and M.O.R.S.; formal analysis, J.E.O.M. and A.C.d.S.M.; investigation, I.M.R.-B. and M.O.R.S.; resources, I.M.R.-B.; data curation, J.E.O.M.; writing—original draft preparation, J.E.O.M. and I.M.R.-B.; writing—review and editing, I.M.R.-B.; visualization, M.O.R.S.; supervision, I.M.R.-B.; project administration, I.M.R.-B.; funding acquisition, I.M.R.-B. All authors have read and agreed to the published version of the manuscript.

**Funding:** This research was funded by the National Council of Science and Technology of Mexico (CONACYT), which financed the project No. 257588, and the scholarship 715424 for Julio Enrique Oney Montalvo and the scholarship 745308 for Adriana Cristina De Silva Madrigal.

**Institutional Review Board Statement:** Not applicable.

**Informed Consent Statement:** Not applicable.

**Data Availability Statement:** All the data is available in the manuscript file.

**Conflicts of Interest:** The authors declare no conflict of interest.

## Appendix A

**Table A1.** Average values of temperature, humidity and light of the greenhouse at the different PTD.

| PTD | Temperature (°C) | Humidity (%) | Light (cd/m$^2$) |
|---|---|---|---|
| 320 | 41.8 ± 2.0 | 87.6 ± 1.7 | 4787.3 ± 320.3 |
| 334 | 38.7 ± 1.3 | 90.1 ± 1.5 | 4223.0 ± 273.0 |

**Table A2.** Parameters measured to ensure a proper ripeness classification of the habanero peppers.

| Ripeness Degree | L | a | b | Humidity (%) | Length (mm) | Width (mm) |
|---|---|---|---|---|---|---|
| Unripe | 43.77 ± 1.72 [b] | −12.0 ± 0.9 [c] | 27.08 ± 3.24 [c] | 85.8 ± 1.1 [a] | 30.8 ± 2.1 [a] | 23.7 ± 2.8 [a] |
| Half ripe | 49.93 ± 4.95 [ab] | 2.4 ± 2.8 [b] | 37.41 ± 3.81 [b] | 81.0 ± 2.3 [b] | 31.3 ± 1.8 [a] | 22.7 ± 1.6 [a] |
| Ripe | 54.19 ± 2.06 [a] | 23.4 ± 4.4 [a] | 43. 9 ± 2.92 [a] | 76.8 ± 4.8 [c] | 32.0 ± 2.0 [a] | 23.2 ± 2.6 [a] |

**Note:** L = Lightness, a = value relative to the green–red color, b = value relative to the blue-yellow color. Different letters in the same column indicate statistically significant differences using least significant difference (LSD) test at $p \leq 0.05$.

## Appendix B

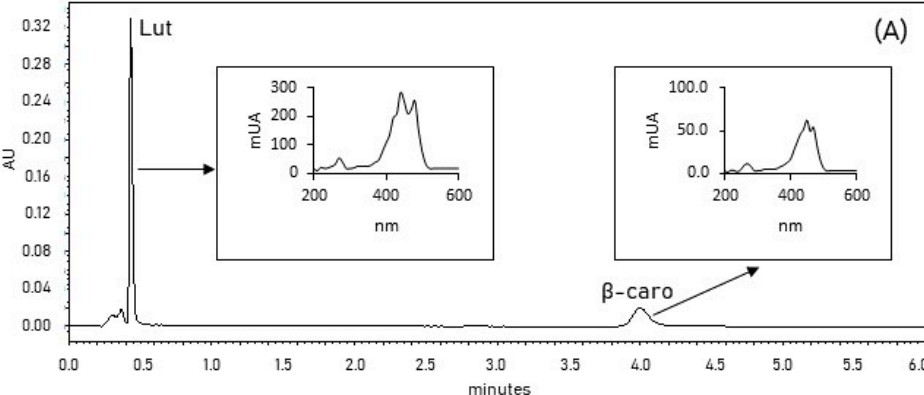

**Figure A1.** *Cont.*

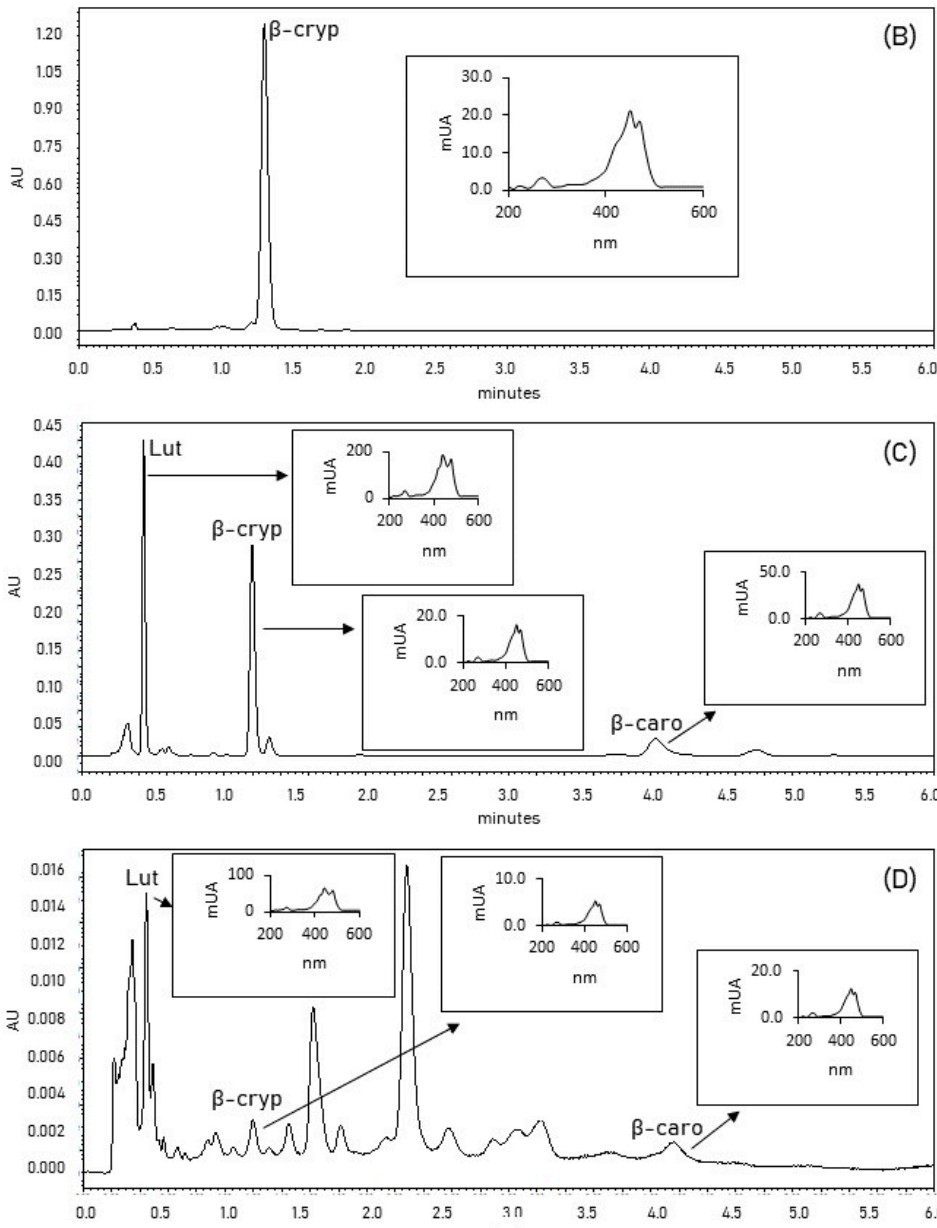

**Figure A1.** Chromatograms of carotenoids at 475 nm: lutein (Lut), β-carotene (β-caro) and β-cryptoxanthin (β-cryp). (**A**) Standards of lutein and β-carotene. (**B**) Standard of β-cryptoxanthin. (**C**) Extracted from unripe (color green) Habanero pepper grown in red soil and harvested at 334 PTD. (**D**). Extracted from ripe Habanero pepper grown in red soil and harvested at 334 PTD. Extraction method: 500 mg of Habanero pepper with 2.5 mL of ethanol was sonicated at 42 KHz for 20 min and then, centrifuged at 3500 rpm for 30 min and filtered through a 0.2 μm PTFE filters.

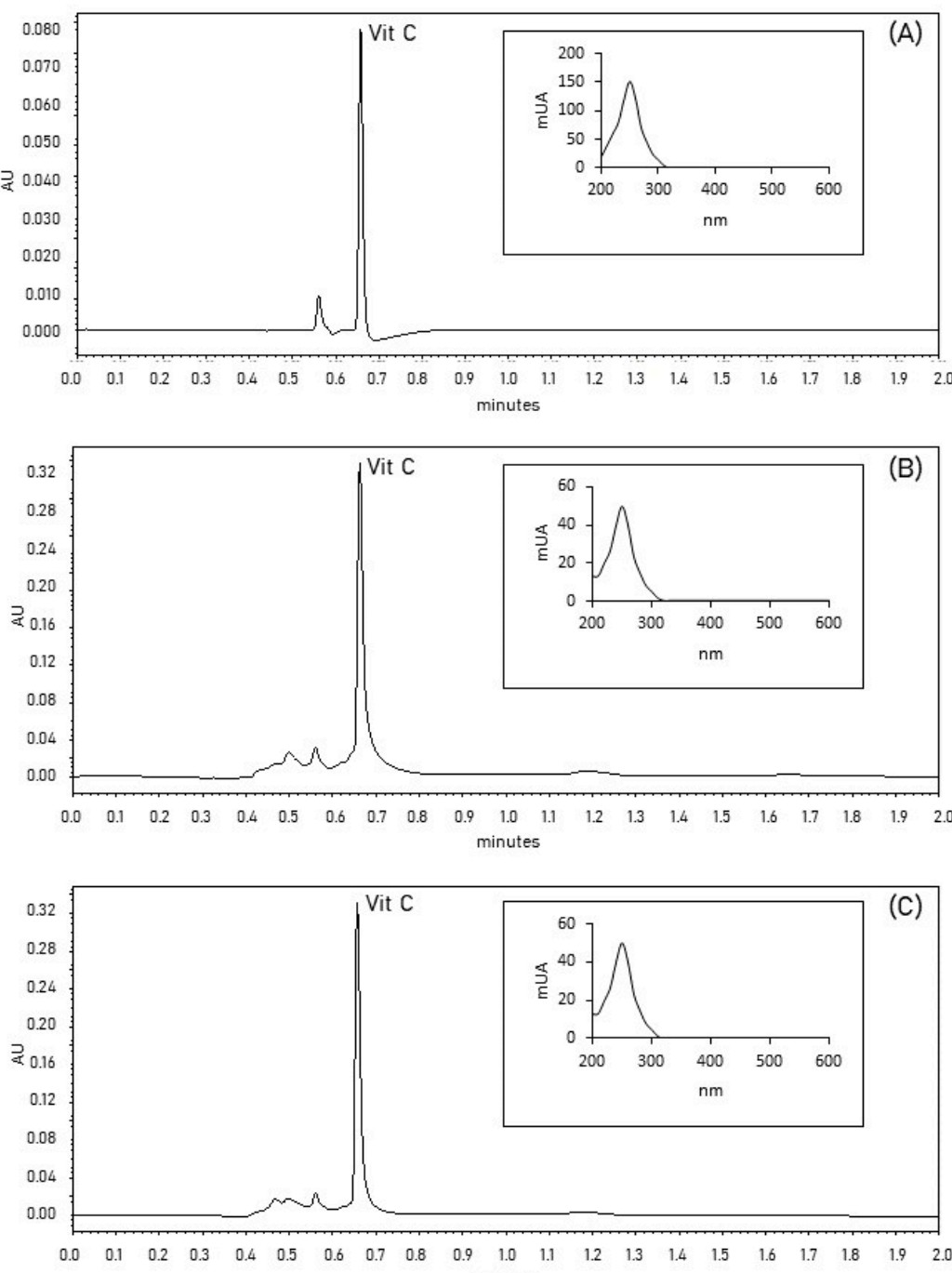

**Figure A2.** Chromatograms of Vitamin C (vit C) at 244 nm. (**A**) Standard. (**B**) Extracted from unripe Habanero pepper grown in red soil and harvested at 320 PTD. (**C**) Extracted from ripe habanero pepper grown in red soil and harvested at 320 PTD. Extraction method: 50 mg of Habanero pepper powder with 4 mL of water:acetonitrile (80:20) was sonicated at 42 KHz for 30 min then, filtered through a 0.2 μm PTFE filters.

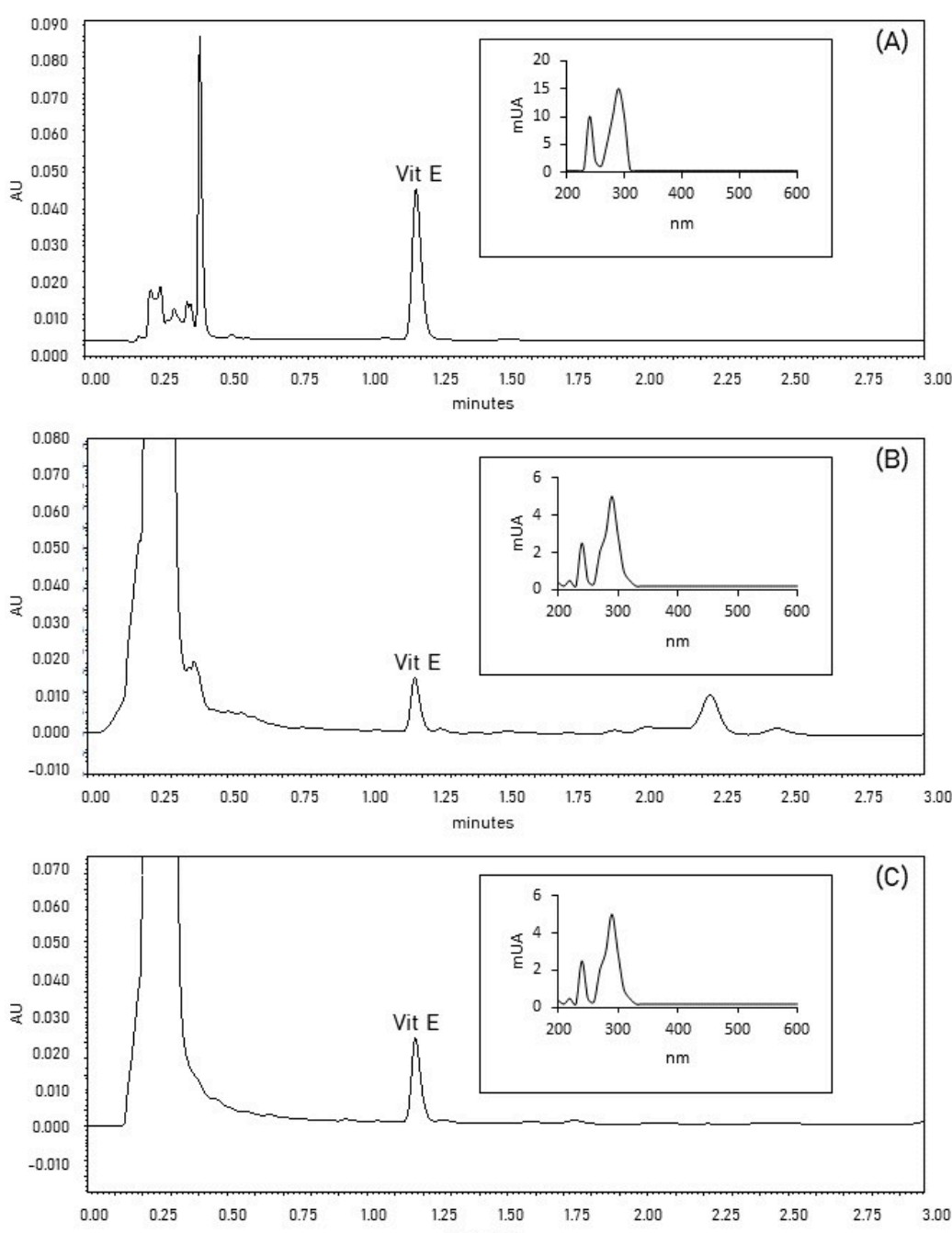

**Figure A3.** Chromatograms of Vitamin E (vit E) at 290 nm. (**A**) Standard. (**B**) Extracted from unripe Habanero pepper grown in black soil and harvested at 320 PTD. (**C**) Extracted from ripe Habanero pepper grown in black soil and harvested at 320 PTD. Extraction method: it was carried out with the same extraction methodology of the carotenoids.

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
