# Peer review of "Effect of the Soil and Ripening Stage in Capsicum chinense var. Jaguar on the Content of Carotenoids and Vitamins"

_horticulturae, doi:10.3390/horticulturae7110442_

Round 1

Reviewer 1 Report

1) This manuscript (Ms) contains misprints, mistakes in English grammar and in the writing style. I recommend that the authors should use some help of a native English speaker or send the Ms to an English Editing Service that proofreads scientific writing. It is very difficult to understand different parts of the manuscript in its present form.

2) Why did the authors analyze only lutein, β-carotene, and β-criptoxanthin carotenoids? And only vitamin E and C? As I known carotenoids of pepper comprise mainly of the unique, powerful and highly stable capsanthin and capsoroubin, together with β-carotene, β-cryptoxanthin, lutein, zeaxanthin, antheraxanthin and violaxanthin. Also, pepper is a rich source of vitamin A. I want to indicate, that Horticulture is journal with high Impact Factor, therefore, authors should improve experimental data presented in Ms text and should improve discussion and conclusions. For example, authors can analyze a complete list of carotenoids or vitamins.

3) Fig. 1. – present bar (5 or 10 cm).

4) “The concentration of vitamins increases as the ripening process is developed, with the highest concentration of vitamin E (9.69±0.02 19 mg/100g of dry mass) and vitamin C (119.44±4.72 mg/100g of dry mass) in the unripe peppers.” In “unripe peppers”? May be in a ripe pepper?

5) Authors should improve the presented HPLC data.

a) present HPLC for carotenoids extracted from unripe and ripe pepper.

b) improve legends for HPLC figures, e.g. explain extraction method, used n.m., and used abbreviations.

c) present HPLC for vitamin C extracted from unripe and ripe pepper.

d) present HPLC for vitamin E extracted from unripe and ripe pepper.

e) how did the authors prove that the peaks shown are the peaks of the compounds described? Chromatograms of standards of pure compounds (lutein, β-carotene, and β-criptoxanthin, vitamin C, E) are usually given.

6) Did I understand correctly that the figures (Fig. 2, 3, 4) are a repetition of the data presented in the tables? If so, it is better to present all the data in the form of histograms (figures), remove the tables. Histograms are perceived more easily.

Reviewer 2 Report

- The authors present the result of a well-organized experiment related to the effect of ripening stage and soil on some quality parameters of habanero peppers (carotenoids and vitamins).

- Why are the harvest conducted at 320 and 334 PTD and not at only one post-transplant day?

- Line 105: Were the two types of tested soils representative of those in the region (for Habanero pepper crop)?

- Line 137: How were the peppers selected?

- Lines 106-108: How were the peppers classified into the three categories (unripe, half ripe and ripe)? Were any ripeness parameters measured?

- It would be very interesting to test other magnitudes related to ripeness stage (size, color (L, a and b), soluble solid content (sugars, ⁰brix),…). The classification between the three ripeness categories should be based on other magnitudes apart than visual color.

- Figure 1 and 2: Some captions are very small

- Peppers are classified as unripe, half ripe and ripe, but in Tables 1 and 2 they are classified as green, green-orange and orange. A unified criteria should be used.

- Table 1 and 2: Differences between the two PTD could be more extensively discussed.

- No significant differences are found in peppers grown in the two different soil types. This fact could be summarized in a graph and discussed. Could it be due to the type of soils used in the experiment? Were the tested soils representative of the different soils in which Habanero peppers are grown?

- No significant differences were found between unripe and half ripe peppers in total carotenoids and antioxidant activity. It would be very interesting to test more than three ripeness categories (intermediate half ripe categories).

- In the abstract and in the conclusions: “The knowledge obtained from this research could be used by the agribusiness to select the best stage of maturation necessary in the habanero pepper for its later use (if a higher content of carotenoids or vitamins is desired) and to provide added value to this fruit and the foods developed from it”. More specific recommendations could be added.

Round 2

Reviewer 1 Report

- Major:

1) “Question 1. This manuscript (Ms) contains misprints, mistakes in English grammar and in the writing style. I recommend that the authors should use some help of a native English speaker or send the Ms to an English Editing Service that proofreads scientific writing. It is very difficult to understand different parts of the manuscript in its present form.”

Answer 1. Thank you for the observation. The grammar and writing style were revised.

-- I do not see a significant English Editing. The Ms requires editing the English language.

2) «b) improve legends for HPLC figures, e.g. explain extraction method, used n.m., and used abbreviations»

-- I did not find remarks on this question. Authors should explain extraction method, used n.m., and used abbreviations in the legends for all HPLC figures.

3) Authors should include information about manufactures of the used standards of pure compounds (lutein, β-carotene, and β-criptoxanthin, vitamin C, E).

4) Also, can the authors provide the UV spectra of peaks of lutein, β-carotene, and β-criptoxanthin, vitamin C, E in the HPLC figures?

5) Authors should increase the font size in the figure A1, A2, A3 (AU, minutes) and decrease the font size in the figure 2, 3, and 4.

6) Authors should check all the statistical treatment that they have presented in the Ms, e.g. I do not understand “a, ab, and c” in Figure 2a. May be a, b, and c?

7) In my opinion it is better to combine some of the figures (Fig. 2, 3, or4), thus reduce the volume of the Ms.
